# Accountable Agents in Software Engineering: An Analysis of Terms of Service and a Research Roadmap

## Abstract

AI coding assistants and autonomous agents are becoming integral to software development workflows, reshaping how code is produced, reviewed, and maintained. While recent research has focused mainly on the capabilities and impacts of productivity of these systems, much less attention has been paid to accountability: who is responsible when agents generate, modify, or recommend code? In practice, accountability is defined through the Terms of Service (ToS) and related policy documents that govern the use of AI-powered development tools.

In this **vision paper**, we present a comparative analysis of the Terms of Service for widely used AI coding assistants and agent-enabled development tools. We examine how these documents allocate ownership, responsibility, liability, and disclosure obligations between tool providers and software developers, and we identify common patterns and divergences between providers. Our analysis reveals a consistent tendency to shift responsibility for correctness, safety, and legal compliance onto users, as well as substantial variation in how providers address issues such as indemnification, data reuse, and acceptable use.

Based on these findings, we argue that existing policy frameworks are poorly aligned with increasingly agent-mediated and autonomous software development workflows. We outline a research roadmap for accountable agents in software engineering, identifying challenges and opportunities for modeling responsibility, designing governance artifacts, developing tooling that supports accountability, and conducting empirical studies of developers' perceptions and practices.

## CCS Concepts

• **Social and professional topics** → **Intellectual property**; **Socio-technical systems**.

## Keywords

AI coding assistants, autonomous agents, accountability, terms of service, governance, liability, intellectual property

**ACM Reference Format:**
Anonymous Author(s). 2026. Accountable Agents in Software Engineering: An Analysis of Terms of Service and a Research Roadmap. In *Proceedings of 3rd ACM International Conference on AI-powered Software (AIware '26)*. ACM, New York, NY, USA, 5 pages. https://doi.org/XXXXXXX.XXXXXXX

## 1 Introduction

AI coding assistants and autonomous agents are increasingly integrated into software development workflows, shaping how code is produced, reviewed, tested, and maintained. Recent work has examined the capabilities of these systems and their impacts on developer productivity and code quality (e.g. [3, 5, 13, 23, 28, 32]). However, as assistants become more agentic (planning and executing multi-step changes, operating over larger code contexts, and acting with reduced direct supervision), questions of accountability become central: who is responsible when agent-generated changes introduce defects, vulnerabilities, or legal risks (e.g., [2, 30])?

In practice, accountability is often articulated less through technical mechanisms than through governance artifacts such as Terms of Service (ToS), product terms, and policy addenda. These documents specify ownership of inputs and outputs, disclaim warranties, allocate responsibilities, restrict acceptable use, and sometimes provide conditional indemnification. Despite their practical importance, ToS are rarely treated as first-class objects of analysis in software engineering research.

In this paper, we analyze 14 policy documents (ToS and related documents) across nine AI coding assistants and agent-enabled development tools. We compare how providers allocate (1) ownership and IP rights, (2) responsibility and liability for correctness, safety, and compliance, (3) data governance (e.g., training and retention), and (4) acceptable use constraints.

Our contributions are as follows: (1) A comparative analysis of how nine providers articulate ownership, responsibility, liability, and data governance in 14 policy documents; (2) a description of recurring policy themes (ownership, responsibility, indemnification, data reuse, acceptable use) and points of variation across providers; and (3) a research roadmap for accountable agents in software engineering.

## 2 Background and Related Work

This section summarizes previous work on AI-assisted and agentic software development, with emphasis on empirical usage evidence, quality and risk considerations, and accountability and governance.

**AI assistants and agentic tooling in software engineering.** Empirical and qualitative research has started to describe how developers integrate code-generating assistants into everyday work. Studies report that interaction is rarely a simple "accept or reject" decision; instead, developers iteratively steer suggestions, inspect generated code, and refactor output to fit local context [3]. At the team and organizational level, assistants have been associated with changes in task completion time and perceived productivity, but also with new needs for coordination and review [5, 23]. More recent work links these workflow changes to concrete development outcomes, including changes in the effects of efficiency, comprehension, and long-term maintenance [6, 22, 25, 28, 32].

In parallel, research has shifted toward *agentic* systems that plan and execute multi-step changes with less direct supervision. These systems are typically evaluated on end-to-end software engineering tasks (e.g., issue resolution) rather than isolated completions [30, 31]. In addition to academic prototypes, practitioner-facing systems and announcements illustrate how quickly this space is moving [12].

**Quality, security, and legal risk.** Although assistants can accelerate development, generated code can be incorrect, insecure, or otherwise unsuitable without careful review. Work on security examines whether AI-assisted coding changes the prevalence and nature of insecure patterns and vulnerabilities, and under what conditions these failures occur [2, 24]. Recent studies discuss the broader risks and limitations of adopting AI-generated code in real projects, including quality trade-offs, maintenance concerns, and how practitioners manage these risks in situ [11, 13].

Legal and compliance risks are also significant. Concerns about training data, memorization, and reuse motivate questions about whether generated code may resemble copyrighted or licensed material, and how similarity across users should be interpreted [4, 17]. Empirical work on memorization and extraction provides grounding for these concerns and highlights mitigation strategies such as dataset deduplication [7, 8, 16].

**Accountability and governance.** As tools become more capable and agentic, accountability becomes a socio-technical problem that spans providers, developers, and organizations. Recent work on AI accountability and governance highlights how responsibility can be distributed (or shifted) between stakeholders, how governance mechanisms shape practical oversight, and what it means to operationalize accountability in deployed systems [21, 26]. Additional perspectives focus specifically on the governance of generative AI [29], the role of records keeping and transparency mechanisms in oversight [9], socio-technical aspects of agentic AI [14], and the fragmented global landscape of AI governance [27]. In addition, documentation artifacts such as datasheets, model cards, and FactSheets aim to clarify intended use, limitations, and responsibilities, thereby supporting auditing and accountability [1, 15, 19].

In this landscape, Terms of Service (ToS), product terms, and policy addenda operationalize accountability choices for AI-powered development tools by defining ownership, responsibilities, disclaimers, and constraints.

## 3 Corpus and method

This study treats Terms of Service (ToS) and closely related contractual documents as governance artifacts that operationalize accountability choices for AI-assisted development tools. Rather than analyzing marketing claims or technical documentation, we focus on binding policy texts that define ownership, responsibility, liability, data use, and constraints on use.

**Selection of tools and documents.** We selected AI-assisted coding tools and agent-enabled development systems that are widely used in practice and/or explicitly support workflows where users can delegate substantial work to an assistant or agent. The resulting corpus covers nine providers and fourteen documents. For each provider, we include the primary contractual documents governing end-user use (e.g., Terms of Use/Service) and, where applicable, product-specific terms, AI-specific service terms, license

agreements, and acceptable use policies. Our goal was not exhaustive legal coverage, but a consistent basis for cross-provider comparison of accountability-relevant clauses.

**Reviewed documents.** Table 1 lists the documents reviewed, including the dates shown by the providers and the URLs last accessed on February 12, 2026. When multiple documents jointly govern an AI feature (e.g., general ToS plus product-specific AI terms), we treated the set of documents as a composite accountability framework for that provider. We report links and dates so that readers can verify the precise versions analyzed, since the policy text can change over time.

**Analysis approach.** We conducted a qualitative, comparative close reading of the corpus with an explicit focus on how accountability is allocated. We coded passages along four recurring dimensions: (1) ownership and intellectual property (inputs and outputs), (2) responsibility and liability (including warranties, disclaimers, accuracy warnings, liability caps, and indemnification), (3) data governance (retention, reuse, and service improvement), and (4) use and delegation framing (including language that anticipates automated or delegated interaction).

Coding was iterative: we first identified recurring clause types (e.g., "you own outputs", "use at your own risk", "no warranties", "indemnify and hold harmless", "liability will not exceed", "we may use content to improve the service"), then compared how each provider combined these clauses into a coherent allocation model. Our intent is not to offer legal advice or doctrinal interpretation, but to analyze how contractual language structures socio-technical accountability in agent-mediated software development.

## 4 Preliminary Findings

In all reviewed documents, we observe a recurring pattern: providers grant users intellectual property rights in the content generated by the system (the "Output"), typically by assigning to users any rights, title and interest that the provider may have in the generated content. In practical terms, this means that users are contractually allowed to use, modify, distribute, and incorporate generated code or text into their own projects. At the same time, these rights grants are paired with clauses that place responsibility for the correctness, legality, and downstream consequences of using that output on the user. However, beyond this shared baseline, differences emerge in the way providers allocate risk, govern data, and frame liability.

**Output rights and user responsibility.** Several providers explicitly assign rights in output to users while simultaneously emphasizing that users bear responsibility for their use. For example, OpenAI states that users "own the Output" and that it "assign[s] to you all our right, title, and interest, if any, in and to Output". At the same time, OpenAI warns that "Output may not always be accurate" and that users "must evaluate Output for accuracy and appropriateness for your use case, including using human review as appropriate". Ownership is thus paired directly with an obligation of independent verification.

Anthropic adopts a similar structure. It provides that "we assign to you all of our right, title, and interest—if any—in Outputs", while prominently cautioning, in capitalized language, that "YOUR USE

**Table 1: Corpus of tools/providers and policy documents reviewed (links as of February 12, 2026).**

| Provider / tool | Document | Date | URL |
|---|---|---|---|
| OpenAI (ChatGPT & API) | Terms of Use | 2026-01-01 | https://openai.com/policies/terms-of-use/ |
| OpenAI (ChatGPT & API) | Service Terms | 2026-01-09 | https://openai.com/policies/service-terms/ |
| GitHub Copilot | GitHub Terms for Additional Products and Features (Copilot section) | 2025-04-01 | https://docs.github.com/en/site-policy/github-terms/github-terms-for-additional-products-and-features |
| GitHub Copilot | GitHub Copilot Product-Specific Terms | 2024-10 | https://github.com/customer-terms/github-copilot-product-specific-terms |
| Anthropic (Claude) | Consumer Terms of Service | 2025-10-08 | https://www.anthropic.com/legal/consumer-terms |
| Anthropic (Claude) | Acceptable Use Policy | 2025-09-15 | https://www.anthropic.com/legal/aup |
| AWS CodeWhisperer | Service Level Agreement | 2023-04-04 | https://aws.amazon.com/codewhisperer/sla/ |
| JetBrains AI Assistant | JetBrains AI Service Terms | 2025-09-30 | https://www.jetbrains.com/legal/docs/terms/jetbrains-ai-service/ |
| JetBrains AI Assistant | JetBrains AI Terms | 2025-05-19 | https://www.jetbrains.com/legal/docs/terms/jetbrains-ai/ |
| Cursor (Anysphere) | Cursor Terms of Service | 2026-01-13 | https://cursor.com/terms-of-service |
| Replit Ghostwriter / AI features | Replit Terms of Service | 2025-08-08 | https://replit.com/terms-of-service |
| Sourcegraph Cody | Sourcegraph AI Terms | 2026-01-22 | https://sourcegraph.com/terms/ai-terms |
| Google (Gemini Code Assist / Gemini API) | Google Terms of Service (Your content) | 2024-05-22 | https://policies.google.com/terms |
| Google (Gemini Code Assist / Gemini API) | Gemini Code Assist Plugin Software License Agreement | 2025-11-05 | https://developers.google.com/gemini-code-assist/resources/plugin-license |

OF THE SERVICES, MATERIALS, AND ACTIONS IS SOLELY AT YOUR OWN RISK".

Replit likewise warns that "Code generated or suggested by our AI systems may be erroneous or incomplete" and that it "accept[s] no responsibility or liability for the accuracy of content on the Service". JetBrains' AI terms clarify ownership boundaries by stating that "You own the Inputs and Your Data", while distinguishing these from "System-generated data [that] includes aggregate anonymized data on how JetBrains AI is used". Ownership of content is preserved, but the responsibility for use remains with the submitting party.

Cursor (Anysphere) allocates responsibility broadly through indemnity language requiring users to "defend and indemnify Anysphere ...from and against any claims, damages, liabilities, losses, and expenses ...arising out of or relating to: (1) your unauthorized use or misuse of the Services; (2) your violation of any applicable laws, regulations or third party rights; or (3) any content submitted by you". The emphasis is less on output ownership and more on downstream accountability. Sourcegraph's AI terms similarly clarify that "As between the parties, you own all Inputs to and Outputs generated by the Service for you", while situating use within broader limitation and liability provisions.

Google's Terms of Service state that users "retain ownership of any intellectual property rights that you hold in that content", embedding AI functionality within a broader platform model in which rights retention coexists with standard service disclaimers. GitHub's Copilot terms provide that users "retain all responsibility for the Suggestions, including for ensuring that they do not violate any applicable law or infringe the intellectual property, privacy, or other rights of any third party" and that, where users enable suggestions matching public code, "you must comply with any applicable third party licenses".

**Risk posture.** Although disclaimers are widespread, providers differ in how directly they allocate downstream legal risk. For example, Replit requires users to "indemnify and hold Replit harmless from any loss or damage incurred by Replit as a result of your use of the platform". Cursor similarly requires users to "defend and indemnify Anysphere" for claims arising out of unauthorized use, legal violations, or submitted content. Both adopt indemnification models that transfer significant litigation exposure to the user.

JetBrains' AI terms require users to "indemnify, defend, and hold JetBrains harmless" for claims relating to "Your Inputs and Outputs or the combination of Your Inputs and Outputs with other data". Anthropic limits its exposure through liability caps stating that total aggregate liability "will not exceed ...the greater of the amount you paid ...and $100". This establishes a quantitative boundary on provider exposure.

OpenAI's Service Terms provide output-related indemnification in enterprise contexts, yet exclude protection where customers "disabled, ignored, or did not use any relevant citation, filtering or safety features". Protection is thus conditional on responsible use of safeguards.

**Data governance.** Another area of divergence is post-submission data use. For example, OpenAI states that it "may use Content to provide, maintain, develop, and improve our Services", embedding improvement rights directly into the contractual framework. Replit connects public content to model development even more explicitly, stating that "Content published in public Apps

may be used by Replit for improving the Service, including but not limited to developing or training large language models".

By contrast, JetBrains distinguishes between user-owned content and analytics data, stating that "System-generated data includes aggregate anonymized data on how JetBrains AI is used", while reaffirming that "You own the Inputs and Your Data". This creates a clearer conceptual boundary between content ownership and system telemetry. Anthropic similarly provides that "as between you and Anthropic … you retain any right, title, and interest that you have in the Inputs", while assigning Outputs to the user.

Google's Terms clarify that users retain ownership of submitted content, while granting Google rights necessary to operate and improve services. The model is platform-centric rather than AI-specific, yet the structural pattern is similar.

**Error framing and bounded remedies.** Providers also vary in how directly they acknowledge AI fallibility and how tightly they constrain remedies. For example, OpenAI states plainly that "Output may not always be accurate". Anthropic provides that the Services and Outputs are delivered on an "AS IS" and "AS AVAILABLE" basis and that use is "SOLELY AT YOUR OWN RISK". Replit warns that the Service "may contain errors, inaccuracies, or omissions" and disclaims responsibility for resulting loss. OpenAI's Beta Services are offered "as-is" and are "excluded from any indemnification obligations", reinforcing the experimental status of certain features.

**Automation awareness and delegated use.** Although most documents assume human oversight, their language is sufficiently broad to encompass automated or delegated interaction. For example, JetBrains provides that users are "solely responsible" for submissions and must ensure they possess all necessary rights to provide inputs. OpenAI prohibits users from representing that Output "was human-generated when it was not", signaling explicit recognition of automated generation contexts.

Anthropic defines Inputs, Outputs, and "Actions" to include "software manipulation" and "system interactions" performed by the Services, thereby anticipating forms of delegated execution.

**Summary.** In summary, across the nine providers, the documents consistently couple output-related rights with user responsibility for correctness and compliance, while diverging in indemnification breadth, data governance structure, and liability caps. Despite increasingly agent-mediated technical capabilities, the contractual architecture remains human-centered: rights may be granted to users, but responsibility remains with them.

## 5 Research roadmap for accountable agents

Accountability in AI-assisted software engineering is currently operationalized primarily through contractual risk allocation: providers grant output-related rights while emphasizing user responsibility, disclaiming warranties, and bounding exposure. This approach is coherent for assistive, human-supervised usage, but becomes increasingly strained as agents plan and execute larger changes with reduced direct supervision. Moving forward, research should focus on aligning accountability mechanisms with agentic reality by advancing responsibility modeling, governance-aware system design, accountability-supporting tooling, and empirical understanding of practice.

**Responsibility modeling in agentic workflows.** The current ToS language tends to treat agent behavior as an extension of user intent, effectively collapsing "delegation" into "use". For agentic workflows, this is an incomplete abstraction: autonomy is gradual, and responsibility may need to be expressed across stages such as planning, proposing, executing, and verifying. Software engineering research can contribute models that represent delegated authority and supervision boundaries in a way that is operationally meaningful. One concrete direction is to connect responsibility to specific artifacts and events in the workflow (e.g., distinguishing responsibility for initiating an action, approving a plan, executing a change, and merging or deploying it) so that accountability is not inferred only after a failure but is represented as part of normal development practice.

**Governance-aware and policy-aware agents.** ToS and related documents function as a primary governance interface between providers and users, but they largely remain external to the technical execution layer: they constrain usage contractually while leaving enforcement to users and organizations. As agents become more capable, there is growing opportunity to make governance constraints computable and actionable. Recent work shows that repository-level context files such as AGENTS.md and CLAUDE.md encode operational guidance, constraints, and project-specific policies in a configuration-like form [10, 20], and that AGENTS.md can improve agent efficiency without degrading task completion while being rapidly adopted across tens of thousands of repositories [18]. These findings suggest a path toward policy-aware agents that can interpret and operationalize governance signals embedded in both contracts and repository artifacts, while preserving meaningful human oversight in delegated workflows.

**Accountability-supporting tooling.** The shift in contractual responsibility assumes that users can validate outputs, but validation becomes harder as automation increases. Technical mechanisms can reduce this gap by making agent actions legible, reviewable, and auditable. In practice, this means tooling that records the provenance of agent-generated changes (including prompts, tool calls, and model/version identifiers), supports reliable audit trails that connect an eventual code state back to the sequence of agent actions, and differentiates review expectations based on origin and risk. These mechanisms do not replace contractual terms; rather, they make it realistic for developers and organizations to satisfy the responsibilities that ToS already impose, particularly when the volume and scope of agent activity exceed what traditional review habits were designed for.

**Empirical studies of practice and perception.** Finally, accountability is not only a policy and tooling question; it is also a question of how people interpret and operationalize responsibility in teams. Empirical research is needed to understand how developers read and internalize the ToS language, whether organizations adjust review and release processes when adopting agentic tools, and how responsibility is negotiated when failures occur. These studies can also examine second-order effects, such as whether explicit contractual disclaimers lead to more conservative tool usage, whether they create a false sense of security when output ownership is emphasized, or whether they push accountability work (testing, auditing, compliance) into roles that are not currently resourced for it.

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
