# OpenReview forum: "Accountable Agents in Software Engineering: An Analysis of Terms of Service and a Research Roadmap"
_ACM.org/AIWare/2026/Conference — AIware 2026_

### Official Review · Reviewer_zQ8z · 2026-03-10

**Rating:** 4
**Confidence:** 4

**Review:**

- I appreciate the motivation and background presented in the paper, which I found clear and well articulated. The topic is timely and highly relevant, and I believe it is worthy of discussion within the community.

- Regarding the method, although this is a vision paper, I would have appreciated a bit more detail about the selection process of the analyzed sources. Of course, this type of paper is not expected to follow extremely rigorous data collection and analysis procedures, so this is not a major issue. However, providing slightly more context about how the sources were selected would strengthen the argument and improve the perceived generalizability of the observations. For instance, the paper refers to “widely used” agents, but it is not entirely clear how these were identified. Were they selected based on existing literature, popularity, usage statistics, or other criteria? A short explanation, possibly supported by references, would make this aspect clearer.

- Another question concerns the quality and adherence of the content found in the analyzed files. This could potentially be discussed in the results section. For example, during the analysis, did the authors observe whether these policy or governance documents actually contain the types of guidance that users would expect? Are these policies clearly written, properly structured, and placed in locations where users can realistically find and use them? Additionally, how readable and understandable are these documents? What level of expertise might be required for users to correctly interpret and apply such regulations? These aspects could open interesting directions for further investigation.

- More broadly, I appreciated the research roadmap presented in the paper, and I agree that most of the proposed directions represent important challenges that the software engineering community will likely need to address in the near future. One potential aspect that could be further considered is the role of inputs and their relationship to outputs. In particular, how can we ensure that input data is handled fairly and responsibly by agents and LLM-based systems? What responsibilities should LLM providers have in controlling or regulating this aspect? More generally, how can the software engineering community study and control the interaction dynamics between humans and agents (especially in terms of contextual communication) rather than focusing only on isolated prompt–response patterns?

- Overall, the paper is well written and clearly presented. I believe it fits well within the scope of the conference and raises several important points for discussion.

**Summary:**

This vision paper presents an analysis of ToS documents for AI Coding agents to understand how providers discuss ownership, responsibilities, and governance mechanisms. Based on these findings, it proposes a research roadmap to ensure accountable use of AI agents in software engineering.

---

> ### Author Response · Authors · 2026-03-21
>
> Thank you for your constructive review.
>
> For the final version of the paper, we will clarify how the analyzed tools were selected. We started from the design-space study of AI coding assistants by Lau and Guo (VL/HCC 2025), which analyzes 90 AI coding assistants across academia and industry, as a starting point for identifying relevant tools:
> https://doi.ieeecomputersociety.org/10.1109/VL-HCC65237.2025.00041
>
> From this list, we attempted to locate publicly available Terms of Service and related policy documents.
>
> However, not all systems listed in that study currently provide accessible governance documentation. Some tools are no longer available, have been merged into other systems, or have been rebranded since the study was conducted. We therefore focused on tools for which publicly accessible Terms of Service or comparable policy documents were available at the time of data collection. We also cross-referenced grey literature on widely used AI coding assistants, such as: https://mstone.ai/blog/top-generative-ai-tools-for-software-development/.
>
> We will clarify this in the paper with wording along the lines of:
>
> “We derived an initial list of AI coding assistants from a design-space analysis of 90 systems in academia and industry (Lau and Guo, 2025). To complement this source and ensure coverage of widely used industry tools, we also cross-referenced practitioner-oriented sources such as the industry overview of generative AI tools for software development (https://mstone.ai/blog/top-generative-ai-tools-for-software-development/). From these sources, we identified tools for which publicly accessible Terms of Service or comparable governance documents were available. Because some systems listed in the design-space study were no longer available, had been merged, or had been rebranded, the final corpus consists of nine tools and fourteen governance documents. While the resulting corpus is not intended to be comprehensive, it captures representative examples of AI coding assistants available at the time of data collection.”
>
> We will also expand the description of the qualitative coding process. Specifically, we will clarify that the analysis was conducted by two researchers who iteratively reviewed the documents and compared policy clauses across providers. During this process, recurring clause types were grouped into four analytical dimensions (ownership/IP, responsibility/liability, data governance, and usage constraints), which were then applied consistently across the corpus. Ambiguous cases were discussed between the researchers until agreement was reached.
>
> The final paper will include a clarification such as:
>
> “The documents were analyzed using qualitative coding across four dimensions (ownership/IP, responsibility/liability, data governance, and usage constraints). Two researchers independently reviewed the documents and iteratively compared policy clauses to identify recurring patterns within these dimensions. Ambiguous cases were discussed until agreement was reached.”
>
> Following the reviewer’s suggestion, we will also briefly expand the discussion to reflect on practical characteristics of these policy documents, such as readability, structure, and the extent to which developers can realistically interpret and apply such governance clauses in practice. In addition, we will extend the research roadmap to explicitly mention the role of input-output responsibility dynamics in agent-mediated development workflows as a potential direction for future research.

---

### Official Review · Reviewer_NxJP · 2026-03-10

**Rating:** 3
**Confidence:** 4

**Review:**

Method and Rigor. The paper analyzes Terms of Service documents from several AI coding assistant providers to understand how responsibility is allocated when AI tools generate code. The approach is reasonable for the research goal, and the authors clearly list the analyzed documents, including links and access dates. However, the sampling strategy is not well justified. The paper states that the tools were selected because they are widely used, but it is not clear how this was determined or whether the corpus is representative of the broader ecosystem. A short explanation of how these providers were identified would strengthen the design. The analysis is described as a qualitative coding of policy clauses related to ownership, responsibility, liability, and data governance. This is appropriate, but the coding process is only briefly described. The paper does not explain whether more than one researcher coded the documents or how the categories were developed and refined. Because of that, it is difficult to assess the reliability of the analysis. In general, the method would benefit from a slightly clearer explanation of how the themes were derived from the documents. Even for a short paper, I think this is a huge fault.

-----

Goal, Results, and Relevance. The goal of the paper is to understand how accountability is defined in AI coding tools through contractual documents. This is a relevant topic, especially given the increasing use of AI agents in software development. The main finding is that most providers grant users rights over generated output but place responsibility for correctness, legality, and downstream use on the user. This observation is supported by examples from several providers. However, the analysis is mostly descriptive; I guess that is due to the nature of the paper (short paper). Yet, the study repeatedly shows that responsibility is shifted to users, but the discussion does not go much deeper than this observation. The research roadmap in the final section is interesting, but it is loosely connected to the empirical analysis. I think the authors could improve the link between the roadmap and what they analysed.

------
Verifiability and Transparency.
The study is transparent about the documents analyzed and provides links to all of them, which is positive. However, the paper does not provide enough detail about the qualitative coding process or examples of how clauses were categorized. Without this information, it is hard to verify the analysis or reproduce it. An explanation of how the coding was conducted would improve transparency.

------
Presentation.
The paper is well written and easy to follow.

-------
Overall Perception.
The topic is interesting and relevant for the AIware community. The idea of studying Terms of Service as governance artifacts in software engineering could open useful research directions. However, the empirical analysis is relatively light and mostly descriptive and the method would be improved with a bit more rigor in the process and description of the process.

**Summary:**

This paper analyzes Terms of Service from several AI coding assistant providers to understand how responsibility, ownership, and liability are allocated when AI tools generate code, and proposes research directions for accountability in agent-mediated software development.

---

> ### Author Response · Authors · 2026-03-21
>
> Thank you for your constructive review.
>
> For the final version of the paper, we will clarify how the analyzed tools were selected. We started from the design-space study of AI coding assistants by Lau and Guo (VL/HCC 2025), which analyzes 90 AI coding assistants across academia and industry, as a starting point for identifying relevant tools:
> https://doi.ieeecomputersociety.org/10.1109/VL-HCC65237.2025.00041
>
> From this list, we attempted to locate publicly available Terms of Service and related policy documents.
>
> However, not all systems listed in that study currently provide accessible governance documentation. Some tools are no longer available, have been merged into other systems, or have been rebranded since the study was conducted. We therefore focused on tools for which publicly accessible Terms of Service or comparable policy documents were available at the time of data collection. We also cross-referenced grey literature on widely used AI coding assistants, such as: https://mstone.ai/blog/top-generative-ai-tools-for-software-development/.
>
> We will clarify this in the paper with wording along the lines of:
>
> “We derived an initial list of AI coding assistants from a design-space analysis of 90 systems in academia and industry (Lau and Guo, 2025). To complement this source and ensure coverage of widely used industry tools, we also cross-referenced practitioner-oriented sources such as the industry overview of generative AI tools for software development (https://mstone.ai/blog/top-generative-ai-tools-for-software-development/). From these sources, we identified tools for which publicly accessible Terms of Service or comparable governance documents were available. Because some systems listed in the design-space study were no longer available, had been merged, or had been rebranded, the final corpus consists of nine tools and fourteen governance documents. While the resulting corpus is not intended to be comprehensive, it captures representative examples of AI coding assistants available at the time of data collection.”
>
> We will also expand the description of the qualitative coding process. Specifically, we will clarify that the analysis was conducted by two researchers who iteratively reviewed the documents and compared policy clauses across providers. During this process, recurring clause types were grouped into four analytical dimensions (ownership/IP, responsibility/liability, data governance, and usage constraints), which were then applied consistently across the corpus. Ambiguous cases were discussed between the researchers until agreement was reached.
>
> The final paper will include a clarification such as:
>
> “The documents were analyzed using qualitative coding across four dimensions (ownership/IP, responsibility/liability, data governance, and usage constraints). Two researchers independently reviewed the documents and iteratively compared policy clauses to identify recurring patterns within these dimensions. Ambiguous cases were discussed until agreement was reached.”
>
> In addition, we will include a short illustrative example in the paper showing how a clause was categorized under one of the analytical dimensions.
>
> Finally, we will strengthen the connection between the empirical observations and the research roadmap by explicitly linking the observed contractual allocation of responsibility to the proposed research directions on responsibility modeling, governance-aware agents, and accountability-supporting tooling.

---

### Official Review · Reviewer_XCMq · 2026-03-11

**Rating:** 2
**Confidence:** 3

**Review:**

+ The framing of ToS documents as governance artifacts is logical and well-motivated.
+ The four coding dimensions (ownership/IP, responsibility/liability, data governance, usage constraints) are analytically clear and well aligned with the research

- Tool Selection Transparency: The paper states that tools were selected (Line 168) because they are “widely used in practice.” It would strengthen methodological transparency to briefly clarify how it was determined. For example, based on Market adoption, GitHub presence, public usage etc. Providing a short sentence would reduce ambiguity in corpus construction.

- Method Description: The qualitative comparative approach is not following any specific guidelines, however the description of coding could be slightly expanded. Such as, multiple authors involved in coding, was there discussion or refinement of codes, any borderline cases encountered etc. are currently missing

- The analysis describes how risk is allocated but remains neutral regarding whether this allocation is appropriate in agentic contexts. The paper could be strengthened by more explicitly addressing on the current contractual architecture is misaligned with increasing autonomy, accountability is meaningfully actionable for developers under current ToS structures etc.

- Quotation Density - Several paragraphs rely heavily on direct quotations from ToS documents. While quotations are necessary for legal precision, extensive use of them interrupt analytical flow. Rewriting clauses in the authors’ own analytical language while retaining short quotations only where wording is needed will improve readability.

**Summary:**

This vision paper analyzes Terms of Service (ToS) and related policy documents for nine AI coding assistants and agent-enabled development tools (14 documents), treating them as governance artifacts that operationalize accountability in AI-assisted software development. Through qualitative comparative analysis, the authors examine how ownership, responsibility, liability, data governance, and acceptable use are allocated across providers. The findings reveal a consistent pattern in providers granting outputrelated rights to users while shifting responsibility for correctness, legality, and compliance to them. Based on these findings, the paper proposes a research roadmap for accountable agents in software engineering, covering responsibility modeling, governance-aware agents, accountability-supporting tooling, and empirical studies of developer practice.

The topic is highly relevant due to the rapid adoption of AI agents in development workflows and the increasing autonomy of such systems.

---

> ### Author Response · Authors · 2026-03-21
>
> Thank you for your constructive review.
>
> For the final version of the paper, we will clarify how the analyzed tools were selected. We started from the design-space study of AI coding assistants by Lau and Guo (VL/HCC 2025), which analyzes 90 AI coding assistants across academia and industry, as a starting point for identifying relevant tools:
> https://doi.ieeecomputersociety.org/10.1109/VL-HCC65237.2025.00041
>
> From this list, we attempted to locate publicly available Terms of Service and related policy documents.
>
> However, not all systems listed in that study currently provide accessible governance documentation. Some tools are no longer available, have been merged into other systems, or have been rebranded since the study was conducted. We therefore focused on tools for which publicly accessible Terms of Service or comparable policy documents were available at the time of data collection. We also cross-referenced grey literature on widely used AI coding assistants, such as: https://mstone.ai/blog/top-generative-ai-tools-for-software-development/.
>
> We will clarify this in the paper with wording along the lines of:
>
> “We derived an initial list of AI coding assistants from a design-space analysis of 90 systems in academia and industry (Lau and Guo, 2025). To complement this source and ensure coverage of widely used industry tools, we also cross-referenced practitioner-oriented sources such as the industry overview of generative AI tools for software development (https://mstone.ai/blog/top-generative-ai-tools-for-software-development/). From these sources, we identified tools for which publicly accessible Terms of Service or comparable governance documents were available. Because some systems listed in the design-space study were no longer available, had been merged, or had been rebranded, the final corpus consists of nine tools and fourteen governance documents. While the resulting corpus is not intended to be comprehensive, it captures representative examples of AI coding assistants available at the time of data collection.”
>
> We will also expand the description of the qualitative coding process. Specifically, we will clarify that the analysis was conducted by two researchers who iteratively reviewed the documents and compared policy clauses across providers. During this process, recurring clause types were grouped into four analytical dimensions (ownership/IP, responsibility/liability, data governance, and usage constraints), which were then applied consistently across the corpus. Ambiguous cases were discussed between the researchers until agreement was reached.
>
> The final paper will include a clarification such as:
>
> “The documents were analyzed using qualitative coding across four dimensions (ownership/IP, responsibility/liability, data governance, and usage constraints). Two researchers independently reviewed the documents and iteratively compared policy clauses to identify recurring patterns within these dimensions. Ambiguous cases were discussed until agreement was reached.”
>
> We will revise parts of the analysis to reduce the density of long verbatim quotations by paraphrasing clauses in analytical language where possible while retaining short quotations where precise legal wording is necessary.
>
> We will also briefly expand the discussion to reflect on whether the contractual allocation of responsibility observed in the analyzed documents may become increasingly misaligned with development workflows as AI coding agents gain higher levels of autonomy. We will highlight this as an open question and identify it as a direction for future research in the roadmap section.